# Combined Multiplexed Phage Display, High-Throughput Sequencing, and Functional Assays as a Platform for Identifying Modulatory VHHs Targeting the FSHR

**DOI:** 10.3390/ijms242115961

**Published:** 2023-11-04

**Authors:** Anielka Zehnaker, Amandine Vallet, Juliette Gourdon, Caterina Sarti, Vinesh Jugnarain, Maya Haj Hassan, Laetitia Mathias, Camille Gauthier, Pauline Raynaud, Thomas Boulo, Linda Beauclair, Yves Bigot, Livio Casarini, Pascale Crépieux, Anne Poupon, Benoît Piégu, Frédéric Jean-Alphonse, Gilles Bruneau, Éric Reiter

**Affiliations:** 1Physiologie de la Reproduction et des Comportements (PRC), Institut National de Recherche Pour l’Agriculture, l’Alimentation et l’Environnement (INRAE), Centre National de la Recherche Scientifique (CNRS), Université de Tours, 37380 Nouzilly, France; anielka.zehnaker@inrae.fr (A.Z.); amandine.vallet@inrae.fr (A.V.); juliette.gourdon@inrae.fr (J.G.); 294515@studenti.unimore.it (C.S.); vinesh.jugnarain@inrae.fr (V.J.); maya.haj-hassan@inrae.fr (M.H.H.); laetitia.mathias@inrae.fr (L.M.); camille.gauthier@inrae.fr (C.G.); pauline.raynaud@inrae.fr (P.R.); thomas.boulo@inrae.fr (T.B.); linda.beauclair@inrae.fr (L.B.); yves.bigot@inrae.fr (Y.B.); livio.casarini@unimore.it (L.C.); pascale.crepieux@inrae.fr (P.C.); anne.poupon@inrae.fr (A.P.); benoit.piegu@inrae.fr (B.P.); frederic.jean-alphonse@inrae.fr (F.J.-A.); 2Unit of Endocrinology, Department of Biomedical, Metabolic and Neural Sciences, University of Modena and Reggio Emilia, 41125 Modena, Italy; 3Inria, Inria Saclay-Ile-de-France, 91120 Palaiseau, France; 4MAbSilico, 1 Impasse du Palais, 37000 Tours, France

**Keywords:** phage display, VHH, GPCR, FSHR, next-generation sequencing

## Abstract

Developing modulatory antibodies against G protein-coupled receptors is challenging. In this study, we targeted the follicle-stimulating hormone receptor (FSHR), a significant regulator of reproduction, with variable domains of heavy chain-only antibodies (VHHs). We built two immune VHH libraries and submitted them to multiplexed phage display approaches. We used next-generation sequencing to identify 34 clusters of specifically enriched sequences that were functionally assessed in a primary screen based on a cAMP response element (CRE)-dependent reporter gene assay. In this assay, 23 VHHs displayed negative or positive modulation of FSH-induced responses, suggesting a high success rate of the multiplexed strategy. We then focused on the largest cluster identified (i.e., PRC1) that displayed positive modulation of FSH action. We demonstrated that PRC1 specifically binds to the human FSHR and human FSHR/FSH complex while potentiating FSH-induced cAMP production and Gs recruitment. We conclude that the improved selection strategy reported here is effective for rapidly identifying functionally active VHHs and could be adapted to target other challenging membrane receptors. This study also led to the identification of PRC1, the first potential positive modulator VHH reported for the human FSHR.

## 1. Introduction

The follicle-stimulating hormone receptor (FSHR) is a class-A G protein-coupled receptor (GPCR), characterised by a large ectodomain that is responsible for the binding of its cognate ligand, the follicle-stimulating hormone (FSH) [1]. In testicular Sertoli cells, FSHR contributes to spermatogenesis [2], whereas, in ovarian granulosa cells, it is involved in follicle maturation and oestrogen synthesis [3].

The FSHR is involved in pathologies affecting the reproductive tract such as ovarian cancers [4], infertility [5], polycystic ovarian syndrome (PCOS) [6,7], or oligozoospermia [8,9,10].

Treatments for assisted reproductive technologies rely on the use of recombinant FSH, as no pharmacological alternative is approved on the market. Recombinant FSH treatment is massively used and is responsible for a list of side effects, including nausea, weight gain, or, more rarely, iatrogenic hyperstimulation syndrome [7]. More recently, paralleling increased circulating levels of FSH occurring at menopause, FSH/FSHR have been reported to likely play a role in highly prevalent pathologies such as obesity, osteoporosis, cardiovascular diseases [11], or Alzheimer’s disease [12,13]. It is, therefore, of great interest to develop novel pharmacological agents capable of modulating FSHR signalling without inducing undesired side effects.

GPCRs are the most abundant family of receptors in the genome and also represent the first class of drug targets [14]. GPCR agonist binds at so-called “orthosteric sites”, which, in the case of FSHR, extend to the extracellular portion of the receptor. This orthosteric ligand binding then induces a change in structure at a distinct spatial region via “allosteric” mechanisms, thereby promoting its ability to couple to transducers, mostly G proteins and β-arrestins. Importantly, some ligands have been shown to bind receptors at sites that are distinct from the orthosteric site. These so-called “allosteric ligands” preserve the spatial and temporal regulations imparted by the physiological ligand which binds at the orthosteric site [15]. Allosteric modulators can either increase the functional response to the orthosteric ligands (i.e., positive allosteric modulators, PAMs) or decrease it (i.e., negative allosteric modulators, NAMs). Neutral or silent allosteric modulators (SAMs) have also been reported [16]. Moreover, ligands of different natures (i.e., small chemicals, peptides, antibodies, etc.) and efficacies can produce distinct population shifts in the natural GPCR conformational repertoire. This conformational selection explains GPCR’s functional selectivity, also known as “biased agonism”, that can modify the balance across the different cellular signalling responses at a given receptor [17].

Paradoxically, despite the tremendous success encountered by antibodies as therapeutic agents, only two GPCR-targeting therapeutic antibodies have been approved to date [18]. Variable domains of heavy chain-only antibodies (VHHs), or nanobodies, are small antibody fragments derived from heavy chain-only antibodies (HcAbs) found in camelids [19]. Their small size (i.e., 12~15 kDa) combined with their unique structural features (i.e., long CDR3, ability to interact with concave surfaces, bind cryptic/less conserved epitopes that full-length antibodies cannot reach) make them interesting tools for challenging targets such as GPCRs [20]. Thanks to these characteristics, VHHs have shown their potential to modulate GPCRs allosterically and have also proven to be instrumental tools in structural biology for this class of receptor [21]. GPCR-specific VHHs are typically obtained from phage-display experiments using immune or synthetic libraries. However, the process of selection and screening allowing the identification of high-affinity/selectivity VHHs remains very challenging, and the attrition rate is even higher when specific pharmacological profiles are sought after.

In the present study, we addressed this issue by combining phage-display multiplexing, next-generation sequencing (NGS), and robust functional screening to generate a panel of VHHs potentially modulating the activity of the FSHR. We further characterised one of the VHHs and found that it acts as a potential PAM for the cAMP pathway activated downstream from the FSHR.

## 2. Results

### 2.1. Multiplexed Phage-Display and NGS Analysis

Two llamas (*Lama glama*) were immunized using different approaches: one was injected with membranes of HEK293 cells overexpressing the human FSHR; the other was injected in the leg muscle with an expression vector encoding for human FSHR complexed with a nanocarrier. Once transferred into muscle cells, the expression vector led to the expression of FSHR at the cell surface, thereby triggering the immune response. In both cases, leukocytes were isolated from blood and their RNAs were purified. After reverse transcription, VHH repertoires were PCR-amplified and cloned into M13 phagemid vectors. The two immune libraries were subsequently used for phage-display experiments. In order to address the challenge posed by the identification of VHHs capable of specifically modulating the activity of the FSHR, we developed an original selection and screening strategy (Figure 1). Briefly, different forms of the human FSHR were used in a total of 12 phage-display experiments as follows: (i) full-length human FSHR was transfected in HEK293 or CHO cells (depending on the panning round), and living cells were used in panning; non-transfected cells served as negative controls; (ii) membranes were prepared from transfected cells; (iii) empty (negative control) or human FSHR-positive proteoliposomes made using cell-free translation; (iv) the extracellular part of the FSHR with or without fusion with maltose-binding protein (MBP) were produced in *E. coli*; MBP was used as a negative control; (v) the N-terminal peptide (AIELRFVLTKLRVI) located in the FSHR ectodomain.

The phage fractions from each round of each panning modality were amplified by PCR and NGS-sequenced. The sequences obtained were aligned and grouped into clusters of identical sequences. Each sequence found enriched in one or more FSHR-positive modalities and not in the negative controls was considered a potential candidate. Applying these criteria, 34 candidate clusters were identified (Table 1). In the prospect of in vivo testing and to benefit from the avidity associated with homodimers, the 34 VHHs were fused in frame with the Hinge-Fc of mouse IgG2a, resulting in VHH-Hinge-Fcs that were cloned into a mammalian expression vector. Importantly, LALAPG mutations were introduced into the Hinge-Fcs in order to block the immune system (i.e., CDC, ADCC) [22].

### 2.2. Autologously Expressed VHH-Hinge-Fc Assayed on CRE-Driven Luciferase Reporter Gene as a Primary Functional Screening

In order to set up a fast and robust functional assay in cells, we assessed the capacity of the different co-transfected VHH-Hinge-Fcs fused to a signal peptide to impact CRE-dependent transcription in cells also co-expressing the FSHR and a CRE-sensitive luciferase reporter gene (Figure 1). The luminescence values shown in Figure 2 are expressed as a percentage of the response obtained with the negative control condition consisting of a non-relevant VHH-Hinge-Fc. The VHH-Hinge-Fcs induced a variety of effects on CRE-dependent transcription in the presence of 3 different FSH concentrations (i.e., 0.3, 1.0, and 3.0 nM). Notably, similar results were observed at the three FSH concentrations. Considering the 0.3 nM concentrations, 8 of the 34 candidates showed a negative effect on the transcription, whereas 15 others, including PRC1-Hinge-Fc, induced an increase in the CRE-dependent transcription when compared to the non-relevant control. In the NGS analyses, the number of sequences per cluster was predictive of VHH affinity for the FSHR. In this functional screen, the data indicate that 23 VHHs exert either negative or positive modulation on FSH-induced activity and are, therefore, likely binding to FSHR. Moreover, some of the nine VHH-Hinge-Fcs that did not display any effect in the CRE–luciferase assay may also bind to the FSHR without eliciting any modulation on its FSH-induced signalling. Therefore, primary screening suggested that the multiplexed phage display strategy succeeded in identifying binders at the FSHR at a very good rate, with a large proportion showing functional modulation on FSH-induced activity.

### 2.3. Clustering of VHH Sequences as a Mean to Assess Epitopic Diversity among Candidate VHHs

We next asked to what extent the selected candidates were related to each other. It is generally admitted that nanobodies with similar VDJ sequences likely bind the same epitope on the target. We therefore adapted the seqUMAP (uniform manifold approximation and projection) that uses kmer sequence embeddings [23] and inserts sequences into two-dimensional (2D) space, such that closely related sequences are neighbours [24]. Figure 3A shows seqUMAP embeddings of the 34 selected candidates in the space created by the complete set of NGS sequences overlaid with different data for each variant. Interestingly, 32 out of the 34 candidates, including PRC1, were clustered together, whereas 2 of the candidates were very clearly unrelated.

To further explore the potential epitopic diversity covered by the selected candidates, we carried out two unrelated yet complementary in silico analyses.

First, we predicted the preferential binding regions on the FSHR ectodomain (ECD) for each candidate using the MAbTope method [25]. Then, the overlaps between all the predicted binding regions were computed and plotted as a heat map in an overlap matrix (Figure 3B). This analysis predicted at least three distinct classes of VHH, each potentially binding a distinct epitope. PRC1 is predicted to belong to one of those three clusters. Next, we carried out a similarity analysis based on a method that we recently reported [26]. In addition to sequence similarities, this method also integrates elements of the 3D structure of the VHH. The rationale is that, if different VHHs are structurally similar, they will likely bind to the same epitope. However, it is important to keep in mind that the opposite is not necessarily true, as different similarity clusters could potentially bind to the same epitope. The similarity matrix has been computed and represented as a heat map (Figure 3C).

Several clusters of similar (ranging from 7 to 2 in size) VHHs were predicted. In this analysis, PRC1 appears to be unique. As expected, the number of clusters was higher in the similarity than in the overlap analysis. We then asked whether or not the predictions made by these two independent in silico methods were congruent. To answer this question, a coherence matrix was computed and revealed a low rate (i.e., 6) of incoherencies, PRC1 not being one of them (Appendix A). Overall, these in silico analyses suggest that the multiplexed phage display strategy is conducive to exploring epitopic diversity. One can speculate that targeting diverse epitopes at a target of interest could represent an asset for generating drugs with pharmacological diversity.

### 2.4. Validation of PRC1-Hinge-Fc Activity Using expiCHO Extracts and cAMP BRET Assay

Of all the potentially interesting VHHs identified in the NGS, functional screening, and epitope prediction, we chose to concentrate on PRC1-Hinge-Fc for further validation characterisation because it displayed a top 3 positive modulation of FSH-induced CRE–luciferase activity while representing, by far, the largest cluster selected, generally a good predictor of favourable affinity.

In order to validate the primary screening data, we implemented a procedure that could be amenable to medium-throughput testing quite rapidly. To ensure the parallelisation and cost-effectiveness of the process, the cDNA encoding for PRC1-Hinge-Fc was cloned in a mammalian expression vector (pcDNA3.1) downstream from an IL-2 signal sequence to ensure extracellular secretion. The construct was then transfected in ExpiCHO cells in small volumes (2 mL of medium per well in a 12-well plate). Medium of ExpiCHO cells transfected with empty pcDNA3.1 was used as a negative control. We then assessed the impact of PRC1-Hinge-Fc on the cAMP production profile by bioluminescence resonance energy transfer (BRET) assay, using HEK293 cells expressing the FSHR and the cAMP BRET sensor (NLuc-Epac-vv), in the presence or absence of FSH, with 10 µL of unpurified ExpiCHO supernatant containing PRC1-Hinge-Fc. Figure 4 shows the kinetic curves of cAMP production expressed as a percentage of the maximum response to FSH. At the three FSH concentrations tested (i.e., 0.3, 3, and 30 nM), PRC1-Hinge-Fc displayed a clear positive modulation compared to the effect of FSH alone at the same concentration. PCR1-Hinge-Fc alone did not affect cAMP production.

### 2.5. PRC1-Hinge-Fc Binding and Kinetic Rate Constants Determination

To further validate PRC1-Hinge-Fc action on FSHR and to explore its pharmacological properties, we produced and purified it in ExpiCHO. Figure 5A shows a Western blot analysis of the purified PRC1 and PRC1-Hinge-Fc. For the latter, two entities were detected in the gel: the most abundant one corresponding to the expected molecular weight of 45 kDa, and the other one, which was detected by the anti-mouse antibody, likely corresponding to a cleavage product in the VHH or between the VHH and the Hinge.

Then, a biolayer interferometry (BLI) assay was carried out to assess PRC1-Hinge-Fc’s ability to interact with the FSHR ECD directly and to determine the kinetic rate constants. Recombinant FSHR ECD bound to FSH was produced, purified, biotinylated, and immobilised on streptavidin sensors. Increasing concentrations of PRC1-Hinge-Fc were then exposed to the sensors to measure the association before being washed to estimate the dissociation (Figure 5B). After double subtraction of NR control-Hinge-Fc and blank, the association and dissociation curve were fitted with a 1:1 model. The estimated k_on_, k_off_, and K_D_ of PRC1-Hinge-Fc were 3.19 × 10^4^ M^−1^, 1.59 × 10^−3^ M^−1^, and 49.9 nM, respectively. Importantly, these data demonstrate the ability of PRC1-Hinge-Fc to bind the FSH-occupied ECD, confirming an allosteric mode of action.

### 2.6. PRC1-Hinge-Fc Binding Specificity and Binding Mode

Next, we assessed the ability of PRC1-Hinge-Fc to bind to the native form of human FSHR, to cross-react with mouse FSHR and closely related human LHCGR as well as mouse LHR and human TSHR. In order to accomplish this, we carried out flow cytometry analyses. PRC1-Hinge-Fc (300 nM) was applied on cells transfected with cDNA coding human or mouse FSHR, human LHCGR, mouse LHR, or human TSHR. Figure 5C shows that PRC1-Hinge-Fc binds to the human FSHR, precluding the use of rodent models to assess in vivo or ex vivo efficacy. In order to gain a better understanding of PRC1-Hinge-Fc binding mode, we carried out a NanoBiT-based FSH binding assay that clearly demonstrated that neither PRC1 (Figure 5D) nor PRC1-Hinge-Fc (Figure 5E) competes with FSH for binding at the receptor, consistent with an allosteric binding mode.

### 2.7. Pharmacological Profile of PRC1-Hinge-Fc

Next, we quantitatively assessed PRC1-Hinge-Fc pharmacological activity on three hallmarks of FSHR signalling: cAMP production, Gs, and β-arrestin recruitment. To this end, three dedicated BRET assays were carried out in living HEK293 cells transiently expressing the hFSHR. First, FSH-induced cAMP responses were compared, with or without the antibody, at different hormone concentrations. Representative kinetics show that PRC1-Hinge-Fc increased cAMP response (Figure 6B), whereas NR control-Hinge-Fc had no effect (Figure 6A). This positive effect of PRC1-Hinge-Fc was confirmed when the whole-dose responses were analysed (Figure 6C, Appendix A). In the presence of PRC1-Hinge-Fc, the EC50 was significantly shifted (*p* = 0.043) from 0.63 ± 2.9 nM in the control to 0.15 ± 0.07 nM with PRC1-Hinge-Fc, whereas the Emax increased by 2.4 times (14.11 ± 1.71 in the controls versus 34.52 ± 6.65 with PRC1-Hinge-Fc, *p* = 0.052). 

Next, we carried out a BRET assay to explore the ability of PRC1-Hinge-Fc to modulate the coupling to Gαs. Therefore, we used the Gs sensor miniGs or mGs to assess its recruitment at the FSHR upon hormone stimulation [27]. As shown in Figure 6E,F, PRC1-Hinge-Fc increased mGs protein recruitment to the FSHR, whereas the NR control-Hinge-Fc had no effect (Figure 6D). In the presence of PRC1-Hinge-Fc, the 50% effective concentration (EC50) was slightly shifted from 8.61 ± 2.41 nM in the control to 2.5 ± 0.81 nM (*p* = 0.037), whereas the Emax was unchanged. A significant increase was observed at 3 nM FSH.

Then, we assessed the effect of PRC1-Hinge-Fc on FSH-induced β-arrestin recruitment to the FSHR using a BRET assay. Even though a slight increase was measured in the presence of PRC1-Hinge-Fc with 3 nM FSH, this did not reach statistical significance and no significant shift was observed in the dose response (Figure 6G–I).

Overall, our data establish that PRC1-Hinge-Fc is a PAM for Gs recruitment and cAMP production at the FSHR. The fact that no such effect was observed on β-arrestin recruitment suggests that PRC1-Hinge-Fc could potentially act as a biased allosteric modulator.

## 3. Discussion

In this study, we presented an improved method for the selection of modulating VHHs against the FSHR, which belongs to the GPCR family, a particularly important yet challenging class of targets for therapeutic antibodies. Our method is based on the combination of multiplexed phage display and NGS. The idea is that diversifying how the target receptor is exposed/conformed during the selection steps increases the likeliness of obtaining VHHs capable of binding to the native form of the receptor. The drawback of this strategy is that it potentially generates an overwhelming number of clones to assess. In order to circumvent this problem and end up with a tractable number of candidates VHHs, we carried out NGS of all the phage display conditions. Indeed, we found out that randomly picking a limited number of clones, as is usual, often leads to false positives while missing interesting clones. Instead, when analysing millions of NGS sequences and applying very stringent criteria (i.e., no hit found across the different negative controls), the total number of candidates dramatically drops. Our data suggest that our strategy leads to identifying hits at a higher rate than with usual VHH selection pipelines.

A limitation of relying on NGS for candidate selection is that it requires gene synthesis before testing. We took this as an opportunity to reformat VHHs into VHH-Hinge-Fc and clone them into a mammalian expression vector in frame with a signal sequence to allow for secretion.

We reasoned that such reformatting could be beneficial for several reasons: (i) VHH-Hinge-Fc being dimeric benefits from avidity, hence having better apparent affinity; (ii) they present much extended pharmacokinetic properties in vivo thanks to the Fc binding to FcRn [28], meaning that the transition from in vitro characterisation to in vivo testing will be seamless; and (iii) the ability of the candidates to be secreted after transfection in mammalian cells was instrumental in the development of an autologous reporter gene assay that allows fast, robust, and parallelisable functional assay in the first line of screening.

A key advantage of this autologous reporter gene assay is to bypass the bioproduc-tion step, which is tedious and costly. From the 34 candidates selected from the NGS, 23 VHHs had apparent modulatory (negative or positive) effects on the CRE-dependent tran-scriptional activity, suggesting that the multiplexed strategy can favour the selection of candidates with pharmacological diversity. A possible drawback of this method is that antibody concentration in media may vary between candidates, precluding the identification of some interesting candidates. Nevertheless, our procedure allows the rapid detection of positive or negative modulators of receptor activity. Also, it allows the selection of candidates that can be produced efficiently and correctly secreted. To further explore the idea that our strategy can select pharmacologically diverse VHHs, we ran three distinct in silico methods reported to predict the epitopic diversity of antibody sequences. We analysed the germinal lineages of the selected candidates (i.e., UMAP projections), predicted the epitopes and their overlaps between the different candidates (i.e., virtual binning), and computed the sequence and structural similarities of the different candidates. Overall, these in silico analyses suggest that the multiplexed phage-display strategy selects VHHs potentially presenting epitopic diversity. Although there are certainly other factors to consider, one can hypothesise that targeting distinct epitopes on a receptor of interest could favour pharmacological diversity.

Here, as an initial step towards validating the multiplexed phage display strategy, we focused on PRC1, which represented the largest sequence cluster by far. Consolidating that choice, a clear functional effect, consistent with a positive modulation of FSHR activity, was found with PRC1-Hinge-Fc in the autologous reporter gene assay and later confirmed using crude ExpiCHO supernatant in a cAMP BRET assay. After that, PRC1-Hinge-Fc was produced and purified, leading to the determination of its binding kinetic rate constants, its binding specificity, and its effect on FSHR signalling. The data demonstrate that PRC1-Hinge-Fc binds specifically to human FSHR with a KD of 49.9 nM. No cross-reactivity was found with the other glycoprotein hormone receptor family members LHCGR and TSHR. Unfortunately, no cross-reactivity with the mouse FSHR was measured, hampering in vivo studies in rodents. Transgenic mice expressing the human FSHR in the FSHR knockout background could help alleviate this important limitation if available in the future. Interestingly, we found that PRC1-Hinge-Fc indifferently binds to the native FSHR without FSH (flow cytometry) and to the FSHR ECD/FSH complex (BLI). We also observed that PRC1-Hinge-Fc modulates the Gs/cAMP pathway only in the presence of FSH and that neither PRC1 nor PRC1-Hinge-Fc interferes with FSH binding to FSHR (NanoBiT, Promega, Madison, WI, USA). Together, these data indicate an allosteric binding mode on the receptor. However, it is important to discriminate allosteric binding from allosteric activity [29,30]. Demonstrating that PRC1-Hinge-Fc exerts positive allosteric modulation will require further pharmacological characterisation and structural evidence.

Based on the recently reported structure of the FSHR in inactive and active conformations [1], the FSH–ECD appears to undergo a large swing between the inactive (parallel to the plasma membrane) and active (perpendicular to the plasma membrane) conformations. Steric clashes with the plasma membrane could potentially prevent FSH binding with the inactive form. In such a scenario, PRC1-Hinge-Fc could potentially modify the equilibrium between inactive and active FSHR conformation, favouring the latter. Such action would facilitate FSH binding and receptor transduction. Of course, further structural studies, including epitope mapping and pharmacological characterisations, will be necessary to test this hypothesis.

In addition, our data revealed no effect of PRC1-Hinge-Fc on FSH-induced β-arrestin 2 recruitment, suggesting that its PAM effect might be biased. Further investigations will be needed to confirm this hypothesis. A study reported binders at the FSHR without any functional evidence being provided [31]. A second, more recent study also reported VHH binding at the FSHR, which inhibited its cAMP response to FSH [32]. To our knowledge, very scarce anti-GPCR VHHs with PAM activity have been reported: VHHs against metabotropic receptors mGluR2 [33] and mGluR4 [34] are the main examples.

FSHR is a key player in the control of reproduction. Women with loss of function of the receptor experience complete infertility, whereas men display impaired fertility due to sperm’s reduced quality. FSHR is also a potential target for non-hormonal contraception and in ovarian cancers. In addition, the FSH/FSHR axis is increasingly reported to be involved in various pathologies occurring in post-menopausal women. For all these reasons, it is essential to develop a panel of pharmacologically active agents capable of specifically targeting the FSHR to modulate its signalling properties, thereby fine-tuning downstream cellular events and physio-pathological processes. Antibodies in general, particularly VHHs, hold fascinating prospects in that regard. The improved selection strategy we report in this paper could speed up the discovery of more pharmacologically active VHHs, not only for the FSHR but also for other challenging membrane targets.

## 4. Materials and Methods

### 4.1. Ligands and Materials

Recombinant FSH was provided by Merck KGaA (Darmstadt, Germany) and diluted in mQ H_2_O at concentrations appropriate for cell stimulation. Ninety-six well white plates from Greiner Bio-one (Courtaboeuf, France) were used. Coelenterazine-H was from Interchim (Montluçon, France). Anti-hen egg lysozyme (HEL) VHH was chosen to design the NR control VHH-Hinge-Fc [35]. Proteoliposomes were obtained from Synthelis (La Tronche, France).

### 4.2. Cell Culture and Transfection

HEK293A cells (ThermoFisher Scientific, Waltham, MA, USA) were cultured in DMEM medium supplemented with 10% (*v/v*) heat-inactivated foetal bovine serum, 100 IU/mL penicillin, and 0.1 mg/mL streptomycin (ThermoFisher Scientific). Cells were transiently transfected in suspension in 96-well plates with Metafectene Pro (BioNTex, München, Germany) according to the manufacturer’s protocol. ExpiCHO cells (kindly provided by Dr Nicolas Aubrey, Tours, France) were cultured in ExpiCHO Expression Medium (ThermoFisher Scientific) at 37 °C, 8% CO_2_, and 120 rpm shaking (CellTron Infors HT, Massy, France).

CHO cells (ATCC, Manassas, VA, USA) were cultured in a complete Ham’s F-12K medium supplemented with 10% (*v/v*) foetal bovine serum and 0.1 mg/mL streptomycin (ThermoFisher Scientific). Cells were transfected using jetOPTIMUS^®^ according to the manufacturer’s protocol (Polyplus-transfection S.A, Illkirch, France).

### 4.3. Llama Libraries

The VHH libraries used in this study were obtained from 2 different llama immunisations. The first immunisation consisted of the injection of HEK293 cell membranes expressing FSHR. The second library was generated from immunising a llama with cDNA encoding for the hFSHR, hLHCGR, or hTSHR (InCellArt, Nantes, France).

### 4.4. Phage Display

Phage display experiments were performed on proteoliposomes (Synthelis, La Tronche, France), whole cells expressing the FSHR [36], recombinant hFSHR ectodomain fused to maltose binding protein, an N-terminal peptide from hFSHR (MyBioSource, San Diego, CA, USA), recombinant hFSHR ectodomain, or cell membrane expressing hFSHR with appropriate negative controls, by adapting previously described protocols [37,38,39].

DNA purified from amplified phages corresponding to each panning experiment were amplified by PCR using VHH-selective primers and sequenced by Next Generation Sequencing (NGS) (Imagif, Gif-sur-Yvette, France). The sequences were then grouped into clusters of identical sequences. The sequences had to be found only in the phage display conditions with FSHR and not in the controls.

The VHH candidates’ sequences were reformatted by adding Hinge-Fc from mouse IgG2a containing the LALAPG mutations to suppress the effector functions [22]. FLAG tag and an IL2 signal sequence were also included. The constructs were codon-optimised for expression in *Cricetulus griseus* (ExpiCHO cells, ThermoFisher Scientific).

The sequences were cloned into the mammalian expression vector pTwist CMV BG WPRE Neo and synthesised by Twist Bioscience (San Francisco, CA, USA). Plasmids were amplified for transfection with NucleoSpin Plasmid Miniprep kit (Macherey-Nagel, Hoerdt, France) according to the manufacturer’s protocol.

### 4.5. NGS Analysis by Germinal Lineage

NGS sequences were processed using a homemade pipeline written in SnakeMake (unpublished) which carried out the trimming of sequence extremities, their fusion, their translation, and their annotations (i.e., FRs and CDRs). The reading frame was determined in the fused sequences by a protein label conserved in the framework 1 (FR1). Sequences were then translated from this frame. After the translation of the sequences, the complementarity-determining regions of the VHHs (CDR) were extracted and concatenated to create the VHH “signature”. The whole CDR sequences were processed with the program seqcollapse (https://forgemia.inra.fr/benoit.piegu/seqcollapse (accessed on 3 July 2023)) to obtain a set of non-redundant sequences and a file describing, for each non-redundant sequence, the number of sequences for each round of selection.

The program seq2kmermat (https://forgemia.inra.fr/benoit.piegu/seq2kmermat (accessed on 3 July 2023) was used to encode the sequences from the set of non-redundant sequences as vectors of kmer. From the resulting matrix, a Uniform Manifold Approximate and Projection (UMAP) 36 was created using the R package uwot 37. The different representations of UMAP projection were created using the following parameters: kmer size—5, number of neighbours—15, number of components—2, metric—Euclidean.

### 4.6. In Silico Prediction of Epitopes

The MAbTope method, described in Bourquard et al. [25], is based on ranking semi-rigid docking poses using an AI-based scoring function. It allows one to predict the epitope and paratope of a given antibody/antigen pair. Comparison of the epitopes (epitope overlap) is made by counting the number of residues predicted to belong to the epitope of two antibodies with very high, high, or medium probability, divided by the maximal number of such residues in the two compared epitopes. Two antibodies are considered in competition if this ratio is higher than 0.3.

For each VHH, the CDR sequences have been delimited using Chotia’s numbering and the secondary structures predicted. The CDRs are then encoded and compared using the method described in Musnier et al. [26]. Two antibodies having similarity higher than 50 have a 0.95 probability of sharing the same epitope; the probability is 0.75 for a score between 30 and 50.

For each pair of VHHs, the incoherence between the similarity and epitope predictions is computed as the difference between the similarity (normalised by min and max values) minus the epitope overlap. Consequently, a value of 1 corresponds to a pair of antibodies having a similarity of 1 (identical) and an overlap of 0 (no overlap of predicted epitopes).

We used the FSHR ECD (isolated from the FSHR structure, PDB ID 8I2H [1]) as the antigen for those analyses.

### 4.7. Autologous Reporter Gene Assay

To determine the capacity of the candidate VHHs to modulate CRE-dependent transcription, HEK293A cells were transfected with human FSHR, each VHH-Hinge-Fc to be tested, and the pSomLuc plasmid expressing the firefly luciferase reporter gene under the control of the CRE of the somatostatin promoter region [40], in DMEM Red-Free medium (ThermoFisher Scientific). After 48 hours, cells were stimulated with FSH for 6 h. Then, the stimulation medium was removed, at which point cells were put in the presence of BrightGlo Substrate (Promega). After a 5 to 10 min incubation at RT, luminescence was measured 5 times (exposure time of 1s/well) in a Mithras LB 943 plate reader (Berthold Technologies GmbH&Co). Values were expressed in relative luciferase activity units (RLU).

### 4.8. Production of VHH-Hinge-Fc in ExpiCHO Mammalian Cell Expression System

On the day of transfection, ExpiCHO cells (ThermoFisher Scientific) were counted with a LUNA-II cell counter (Logos Biosystems, Villeneuve d’Ascq, France) and density was adjusted to 6 × 10^6^ cells/mL. Transfections were performed in 1.5 mL ExpiCHO Expression medium in 12-well culture plates (ThermoFisher Scientific), with 0.8 mg/mL of plasmid corresponding to VHH-Hinge-Fc per condition according to the manufacturer’s protocol (ThermoFisher Scientific). The culture was incubated at 37 °C, 8% CO_2_, and 120 rpm shaking (CellTron, Infors HT, Massy, France). Sixteen hours after transfection, 5 µL of ExpiCHO Enhancer and 240 µL ExpiCHO Feed (ThermoFisher Scientific) was added per well before putting back the plates at 37 °C, 8% CO_2_, and 120 rpm shaking. Five days after transfection, 240 µL ExpiCHO Feed was added again in each well and plates were incubated at 32 °C, 5% CO_2_, with 120 rpm shaking. After 11 to 12 days of transfection, the medium was collected and centrifuged at 4000× *g*, at 4 °C for 20 min. The supernatants were then collected and stored at –20 °C until further use. The presence of secreted, soluble full-length VHH-Hinge-Fc in the supernatants was confirmed by anti-mouse Western blot.

For larger-scale production, ExpiCHO cells were transfected in 30 mL cultures. Ten days after transfection, cell cultures were centrifuged at 4000× *g*, 4 °C for 20 min, and the supernatant was dialysed against Tris–NaCl buffer (50 mM Tris–HCl pH 8.0 and 100 mM of NaCl) overnight at 4 °C using 6–8 kDa cellulose membrane tubing (ThermoFisher Scientific). The VHH-Hinge-Fc present in the dialysed supernatant was purified through the capture of the protein A domain present in the Hinge-Fc domain, using UNOsphere SUPrA Affinity Chromatography Media (Bio-Rad Laboratories, Hercules, CA, USA). The protein A beads mixed with the supernatant were incubated for 1 h at room temperature on a rotary wheel. The mixture was then centrifuged and the pellet containing the VHH-Hinge-Fc bound to the beads was put on Poly-Prep^®^ Chromatography Column (Bio-Rad Laboratories). The column was then washed extensively with a washing buffer (50 mM Tris–HCl pH 8.0 and 100 mM of NaCl). The VHH-Hinge-Fc was eluted with a solution of 100 mM of glycine at pH 2.9 and neutralised with a solution of 1 M Tris–HCl at pH 8.0. The eluates were then dialysed against 50 mM Tris–HCl pH 8.0 and 100 mM NaCl. The presence of purified VHH-Hinge-Fc was confirmed by SDS PAGE gel and the preparation was stored at −20 °C until further use.

### 4.9. SDS-PAGE

Samples were boiled for 10 min at 90 °C in Laemmli buffer. Proteins were separated by SDS-PAGE on 15% polyacrylamide gels that were stained with Coomassie blue.

### 4.10. BRET Assays

To assess the cAMP production in HEK293 cells, cells were transfected with plasmids coding for human FSHR and an EPAC sensor fused to a nanoluciferase and a tandem acceptor protein, consisting of a circularly permutated Venus and a Venus (Nluc-EPAC-vv, kindly provided by Dr Kirill Martemyanov) [32].

To evaluate Gs protein recruitment, HEK293 cells were transiently transfected with hFSHR C-terminally fused with donor Rluc8 (FSHR-Rluc8) and the miniGs sensor acceptor NES-Venus-mGs (kindly provided by Pr. N.A. Lambert, Augusta University, Augusta, GA, USA) [27,41].

To measure β-arrestin recruitment, HEK293 cells were co-transfected with FSHR-Rluc8 and a β-arrestin-2 N-terminally fused to a YPET BRET acceptor (kindly provided by Dr M.G. Scott, Cochin Institute, Paris, France).

Forty-eight hours after transfection, cells were stimulated with increasing concentrations of FSH (10^−9.5^ to 10^−6.5^ M), 2 µL/well of crude ExpiCHO medium containing the VHH-Hinge-Fc or 1µM of purified VHH-Hinge-Fc diluted in PBS with/containing 5 µM coelenterazine-H. BRET signals were recorded every 60 s for about 60 min in a Mithras LB 943 plate reader (Berthold Technologies GmbH&Co).

### 4.11. Flow Cytometry

To assess the binding of the candidate VHH-Hinge-Fc to FSHR, CHO cells were transfected with hFSHR, mFSHR, mLHCGR, and hTSHR, which were all synthesised (Twist BioSciences), or hLHCGR (kindly provided by Pr. A. Hanyaloglu, Imperial College, London). All the receptors had an N-terminal FLAG-tag except hTSHR. Twenty-four hours after transfection, cells were washed and detached with PBS 10 mM EDTA. The cells were mixed with the diluted VHH-Hinge-Fc in a staining buffer (PBS, BSA 0.5%) and incubated for an hour at 4 °C. After incubation, cells were washed twice with staining buffer and centrifuged 5 min, 500× *g* at 4 °C. Then, cells were stained with a mixture containing an anti-FLAG-PE, anti-6xHis tag-APC (both provided by Miltenyi Biotec Bergisch Gladbach, Germany), and a live-dead dye (Invitrogen, Waltham, MA, USA) diluted in staining buffer. Cells were incubated for 30 min at 4 °C and washed twice with staining buffer. The samples were resuspended in PBS and analysed with a MACSQuant Analyser 10 flow cytometer (Miltenyi Biotec). Data were analysed and plotted with FlowJo 7.6.3 (FlowJo, Ashland, OH, USA).

### 4.12. Binding Validation with BioLayer Interferometry (BLI)

FSHR ECD-Hinge-Fc was produced in ExpiCHO in the presence of FSH and purified according to the protocol described above. The ectodomain was biotinylated using the EZ-link Sulfo-NHS-Biotinylation kit instructions (ThermoFisher Scientific) with a 1:1 biotin:ectodomain *ratio*. The ECD was loaded at a concentration of 10 nM for 240 s on streptavidin-coated biosensors (Sartorius, Aubagne, France) before 2 washing steps with PBS 1X-BSA 0.002%. Then, the sensors were incubated for 100 s with increasing concentrations of PRC1-Hinge-Fc and the dissociation was performed in PBS 1X-BSA 0.002% for 150 s. The association/dissociation profiles and the kinetics constants were calculated with an Octet 96 RED interferometer (Pall Forte Bio, Fremont, CA, USA), and the results were analysed with the Octet Data Analysis software (version 9.0).

### 4.13. NanoBiT FSH Binding Assay

To determine if PRC1-Hinge-Fc competed with FSH binding on FSHR, a luminescence assay based on a split luciferase fused to proteins of interest was conducted [42]. FSH was fused to the LgBit subunit of the luciferase and produced in ExpiCHO cells. FSHR was fused to SmBit and transfected in HEK293A cells. After 48 h, cells were stimulated with 10 nM of FSH–LgBit and either PRC1, PRC1-Hinge-Fc, or the corresponding NR control, and 5 µM of coelenterazine H diluted in PBS was added. Luminescence signals were measured for 65 min in a Mithras LB 943 plate reader (Berthold Technologies GmbH&Co).

### 4.14. Statistical Analysis

The results are shown as mean ± SEM of four or five independent experiments, each performed in triplicate. Data were analysed and plotted using GraphPad Prism 6 (San Diego, CA, USA).

The values of all replicates were expressed as the percentage of the maximum of FSH-induced response. Statistical significance was determined by a two-way ANOVA. *p*-values were considered significant at <0.05.

## Figures and Tables

**Figure 1 ijms-24-15961-f001:**
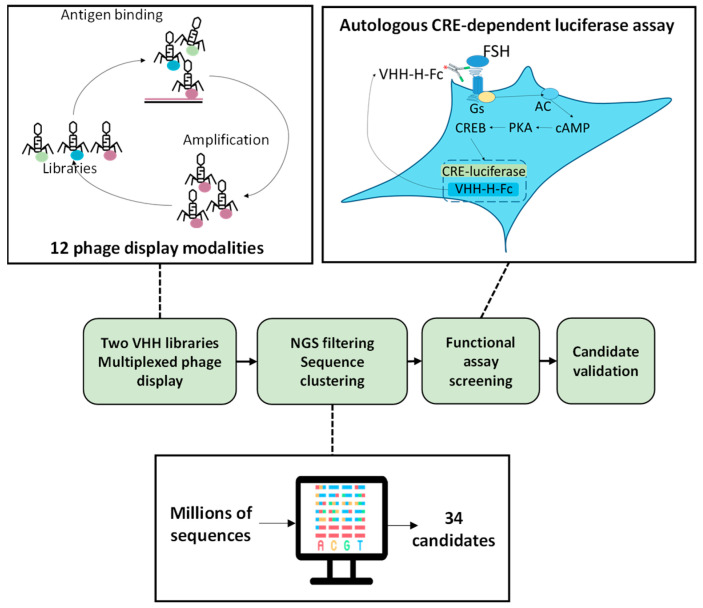
Schematic representation of the strategy used to select anti-FSHR VHH candidates. VHH libraries were constituted from the immune repertoire of two immunised llamas. Libraries underwent multiplexed phage displays involving a variety of panning strategies. Phages recovered from these independent panning modalities were NGS-sequenced, and the sequence analysis led to the identification of 34 clusters of identical sequences of interest. An autologous assay was established based on the co-transfection of each candidate VHH-Hinge-Fc with the FSHR and a CRE-driven luciferase reporter gene. Via the addition of a signal peptide, the VHH-Hinge-Fcs were secreted in the medium where they could autologously target the plasma membrane-expressed FSHR. Luciferase activity was assessed after the cells were exposed to FSH.

**Figure 2 ijms-24-15961-f002:**
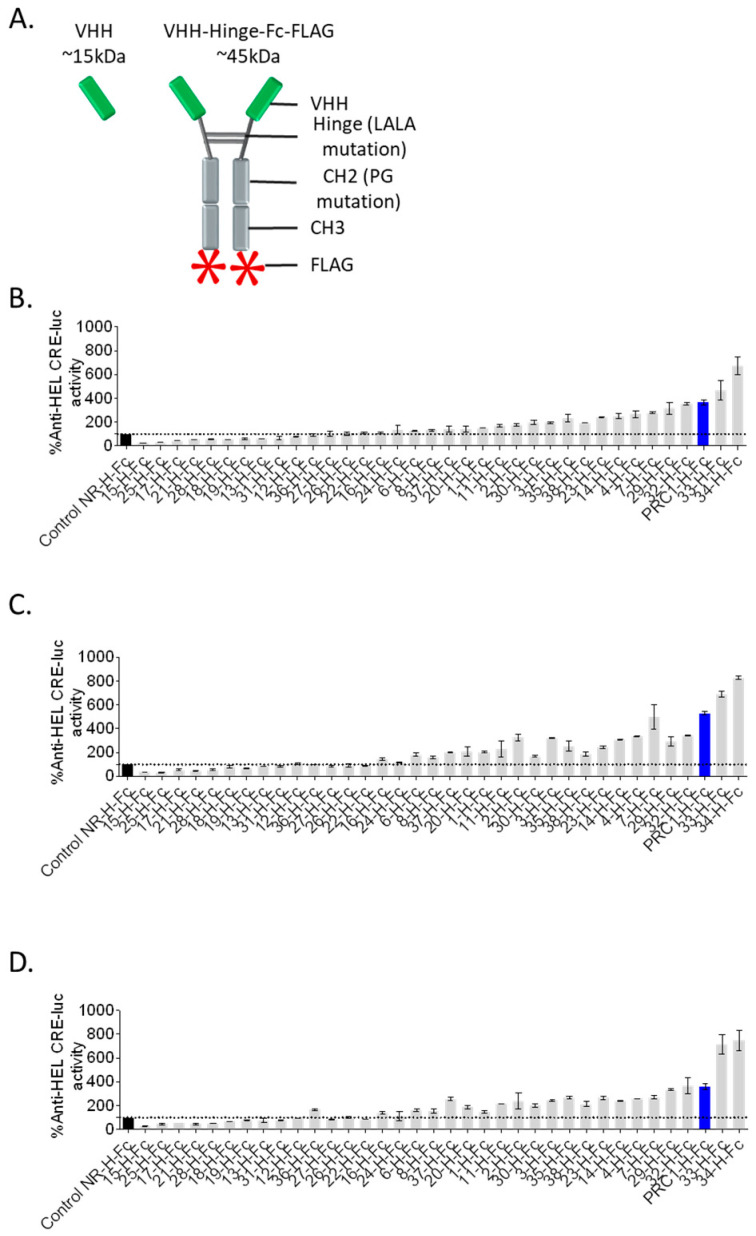
Rapid and robust screening of candidate VHHs. (**A**) The 34 best candidate VHHs were synthesised in VHH-Hinge-Fc format and tested in an autologous functional assay. (**B**–**D**) HEK293 cells were co-transfected with the human FSHR, the reporter gene, and the VHH-Hinge-Fc plasmids. Cells were stimulated with 0.3 nM (**B**), 1.0 nM (**C**), or 3.0 nM (**D**) of FSH for 6 h. The luminescence was measured for each well and then normalised by the maximum FSH response percentage in the presence of the non-relevant control VHH (NR control-H-Fc, black histogram). PRC1-Hinge-Fc (PRC1-H-Fc), which represented the largest sequence cluster, was in the top 3 regarding positive modulation in this assay and was selected for further analyses (blue histogram). Means ± SD of duplicate measurements.

**Figure 3 ijms-24-15961-f003:**
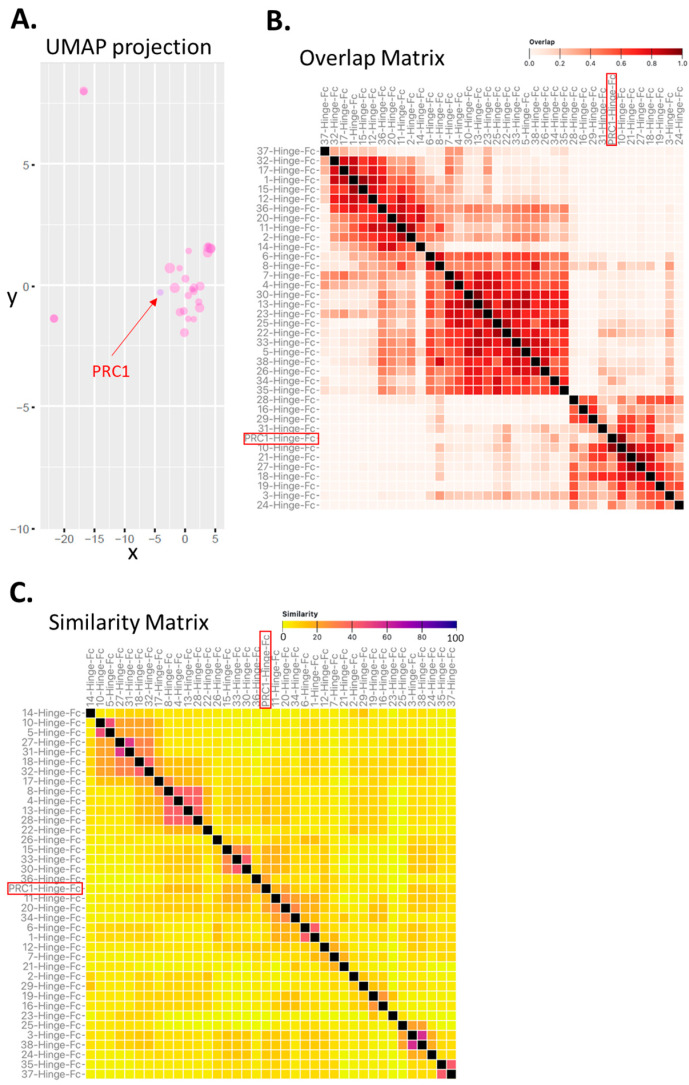
Predicted epitope diversity among candidate VHH-Hinge-Fcs. (**A**) Clustering based on UMAP projection of CDRs from the 34 enriched sequences, considering the UMAP projection obtained with the complete set of NGS sequences as the reference. (**B**) Clustering matrix based on epitope overlap (virtual binning) prediction generated by the MAbTope method. (**C**) Clustering matrix representing sequence and predicted 3D structure similarities in CDRs generated with the MAbCross algorithm. PRC1-Hinge-Fc is indicated with an arrow or boxes in red. Each method predicts clusters that are indicative of potential epitopes.

**Figure 4 ijms-24-15961-f004:**
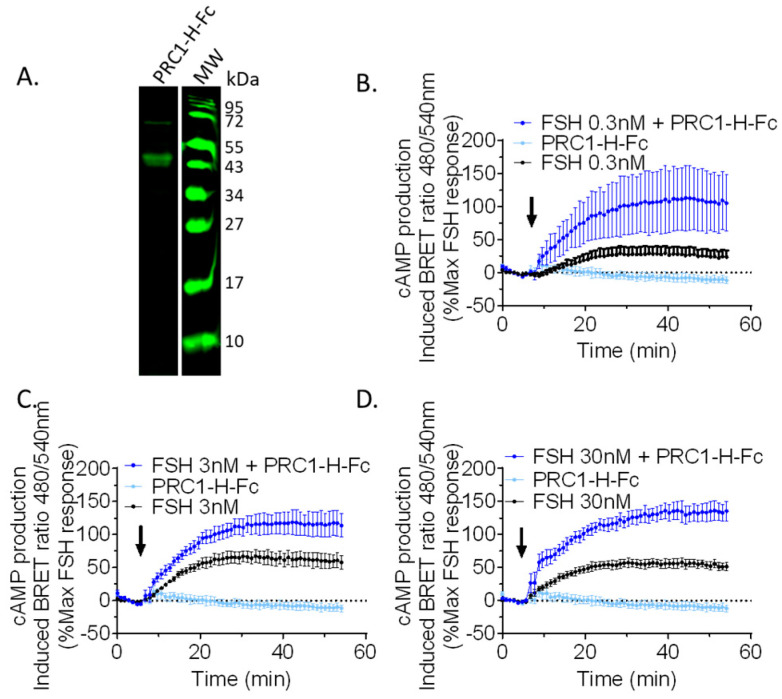
Fast and effective functional validation using crude ExpiCHO cell supernatant and cAMP BRET assay. (**A**) PRC1-Hinge-Fc (PRC1-H-Fc) was produced in ExpiCHO and the medium supernatant was separated on a 15% reducing SDS-PAGE, transferred, and analysed in Western blotting. PRC1-H-Fc was revealed with an 800 nm-dye-conjugated anti-mouse secondary antibody (green). (**B**–**D**) Transfected HEK293 cells were stimulated with 0.3 (**B**), 3 (**C**), or 30 nM (**D**) FSH and 10 µL/well of crude ExpiCHO supernatant, containing or not containing PRC1-Hinge-Fc. The arrows represent the addition time of the FSH and crude supernatant to the cells. Signals were monitored in living cells for 55 min and BRET *ratios* were normalised as a percentage of the maximum FSH response. Data represent SEM of 3–5 independent experiments.

**Figure 5 ijms-24-15961-f005:**
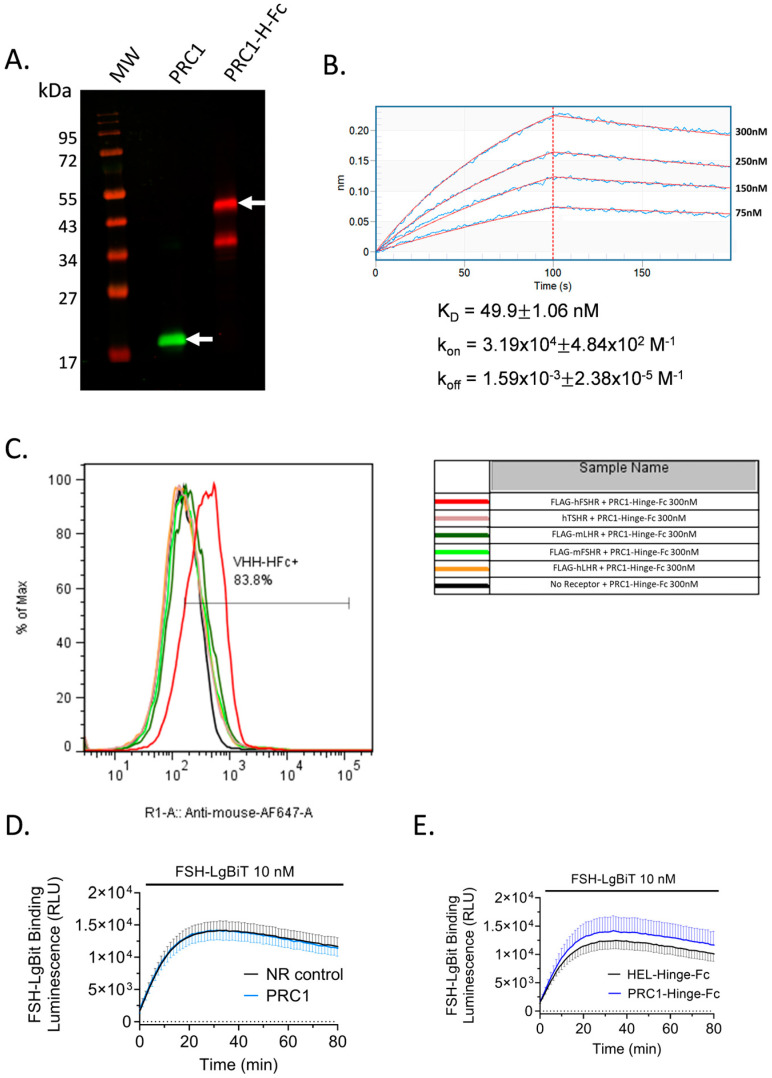
PRC1-Hinge-Fc purification, kinetic rate constants, and binding specificity. (**A**) PRC1-Hinge-Fc was produced in ExpiCHO and purified on protein A column. PRC1 was produced in *E. coli* and purified on Ni-NTA. Both preparations were separated on a 15% reducing SDS-PAGE, transferred, and analysed in Western blotting. PRC1 was revealed with anti-His antibody and an 800 nm-dye-conjugated secondary antibody (green). PRC1-Hinge-Fc was revealed with an anti-mouse antibody conjugated to a 700 nm dye (red). PRC1 and PRC1-Hinge-Fc are indicated with the arrows. (**B**) Biolayer interferometry (BLI) was carried out to determine the kinetic rate constants of PRC1-Hinge-Fc binding to the FSHR ECD/FSH complex. Streptavidin sensors were coated with the biotinylated FSHR ECD-Hinge-Fc and incubated with PRC1-Hinge-Fc at different concentrations. Data were fitted with a 1:1 model. K_D_, k_on_, and k_off_ values are indicated on the graph. (**C**) Binding specificity assessed by flow cytometry. CHO cells transfected with cDNA coding for different closely related GPCRs (hFSHR, mFSHR, hTSHR, mLHCGR, hLHCGR) were incubated with PRC1-Hinge-Fc. PRC1-Hinge-Fc specifically detected hFSHR. (**D**) Competition assay of PRC1 for FSH binding at the FSHR. SmBiT subunit of split nano-luciferase fused in the N-terminus of human FSHR was transfected in HEK293 cells. Forty-eight hours post-transfection, cells were incubated with 10 nM of purified FSH-LgBiT and 300 nM of PRC1 (light blue) or anti-HEL VHH used as an NR control-Hinge-Fc (black). (**E**) Competition assay of PRC1-Hinge-Fc for FSH binding at the FSHR. Cells were incubated with 10 nM of purified FSH-LgBiT and 100 nM of PRC1-Hinge-Fc (dark blue) or anti-HEL-Hinge-Fc used as an NR control (black). (**D**,**E**) Means ± SEM of 2 independent triplicate experiments.

**Figure 6 ijms-24-15961-f006:**
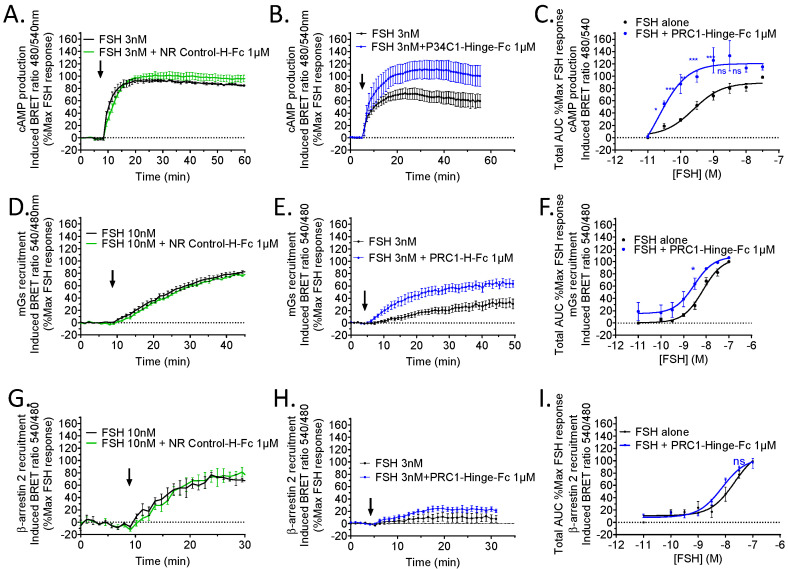
PRC1-Hinge-Fc pharmacological profiling. HEK293 cells expressing hFSHR were stimulated (black arrows) with various concentrations of FSH and 1µM of PRC1-Hinge-Fc or 1 µM of NR control-Hinge-Fc. For each experiment, BRET ratios were normalised as a percentage of the maximum FSH response, and the area under the curves (AUC) were calculated and plotted. In black: FSH stimulation; in blue: FSH + PRC1-Hinge-Fc 1 µM stimulation; in green: FSH + NR control-Hinge-Fc. (**A**) Kinetics of cAMP production in the presence or absence of NR control-Hinge-Fc with 3 nM FSH (N = 3). (**B**) Kinetics of cAMP production in the presence or absence of PRC1-Hinge-Fc with 3 nM FSH (N = 4). (**C**) Dose–response plot from the AUC of cAMP production kinetic curves (N = 4). (**D**) Kinetics of mGs recruitment in the presence or absence of NR control-Hinge-Fc at 10 nM FSH (N = 3). (**E**) Kinetics of mGs recruitment in the presence or absence of PRC1-Hinge-Fc at 3 nM FSH (N = 4). (**F**) Dose–response plot from the AUC of mGs recruitment kinetic curves (N = 4). (**G**) Kinetics of β-arrestin recruitment in the presence or absence of NR control-Hinge-Fc at 10 nM FSH (N = 3). (**H**) Kinetics of β-arrestin recruitment in the presence or absence of PRC1-Hinge-Fc at 3 nM FSH (N = 5). (**I**) Dose–response plot from the AUC of β-arrestin recruitment kinetic curves (N = 5). Data represent SEM of 3–5 independent experiments (* *p* < 0.05, ** *p* < 0.01, *** *p* < 0.001, ns not significant).

**Table 1 ijms-24-15961-t001:** Selected clusters across the different panning modalities. Phages were selected against proteoliposomes, cells, maltose-binding protein (MBP) fusion ectodomain (ECD), peptide ectodomain, or cell membranes containing or not containing (negative controls in red) the human FSHR. Two (R2) or three (no indication or R3) rounds of panning were carried out. The number of sequences was normalised per million for each condition. The number of identical sequences per cluster is indicated. Light grey specifies the smaller clusters, whereas red indicates the larger clusters. Of note, with 45,099 sequences selected, PRC1 represented by far the largest cluster.

	Proteoliposomes		Cells			Maltose-Binding Protein		FSHR Peptide	FSHR Ectodomain		Cell Membranes		
	Control	FSHR (R3)	Control	FSHR (R2)	FSHR (R3)	Control	FSHR (R2)	FSHR (R3)	FSHR (R2)	FSHR (R3)	Control	FSHR (R3)	Total
PRC1									28,890	16,209			45,099
1		200						952	68	34			228
6								560					103
11		114		49			1464	500	54				239
15								451					83
19										757			403
23										447			238
27							128			282			160
31										254			135
2										197			105
7		2074							249				269
12		1313							230				185
16		171					12018			19			964
35		114					8243			26			668
20		95					6715						533
24							4198						327
36							3749		191				331
28				59	6364								480
38					4256								313
32				504	2529								288
3					2271								167
8				178	1972								181
13				405	1822								216
17					1741								128
21				1846	1469							92	493
25				884	1278								273
29				592	163			103					151
33				504	340								127
4				494	476								135
34												776	93
30				20									4
14												451	54
22												42	5
37							2863						223

## Data Availability

Data are contained within the article.

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
