# Peer review of "Combined Multiplexed Phage Display, High-Throughput Sequencing, and Functional Assays as a Platform for Identifying Modulatory VHHs Targeting the FSHR"

_ijms, 2023, doi:10.3390/ijms242115961_

Round 1

Reviewer 1 Report

Comments and Suggestions for Authors

The authors presented an improved and interesting method for selecting VHH against the FSHR, which belongs to the GPCR family, targeting this receptor using VHH-hinge-Fc. The combination of multiplexed phage display, NGS, bioassays, and biochemical characterization of PRC1-Hinge-Fc are well developed. However, some points need to be addressed to improve the manuscript. 

  • It needs to be clarified why the authors chose the PRC1-Hinge-Fc since great levels of cre-luciferase activity were achieved with VHH 33 and 34.
  • The number of clusters was used to select, but it did not correlate with in vitro luciferase assay since 8 out of the 34 candidates showed a negative effect on the transcription. Could authors explain? 
  • Figure 2: “The same screen was carried out at different FSH concentrations with similar outcome (not shown).” Please add data to supplemental materials. Also, the bar plots should have a standard deviation of the assays.
  • The presence of cleavage products is a major concern since it is impossible to determine if the expected protein or the cleavage product performs the binding to the receptor. Additional purification was performed ? If yes, insert the dates.
  • Figure 5 A: The figure should contain the molecular weight of all standard bands.
  • Figure 6: Scales must be the same for the graphs.
  • Discussion must be improved; the authors mostly summarize the results.

Comments on the Quality of English Language

 Minor editing of English language required

Author Response

Reviewer 1’s queries:

  • It needs to be clarified why the authors chose the PRC1-Hinge-Fc since great levels of cre-luciferase activity were achieved with VHH 33 and 34.

We chose PRC1-Hinge-Fc because it represented by far the most enriched cluster with 28890 /106 sequences after round 2 and 16209/106 after round 3 while still being in the top 3 in terms of cre-luciferase activity. We interpreted this high level of enrichment as being predictive of good affinity. A clarification has by added in the revised manuscript (lines 210-214).

  • The number of clusters was used to select, but it did not correlate with in vitro luciferase assay since 8 out of the 34 candidates showed a negative effect on the transcription. Could authors explain?

The number of sequences per cluster is indicative of VHH affinity for the FSHR, not of positive or negative modulatory activity. In principle, negative modulators also need to bind FSHR to exert their functional effects. Moreover, some of the 9 VHH-Hinge-Fc that did not display any effect in the cre-luciferase assay, may actually bind to FSHR without eliciting any modulation on its FSH-induced signaling. These considerations are now explained in the revised manuscript (lines 148-153).

  • Figure 2: “The same screen was carried out at different FSH concentrations with similar outcome (not shown).” Please add data to supplemental materials. Also, the bar plots should have a standard deviation of the assays.

The 3 FSH concentrations with means  SD are now shown is the revised Figure 2. The result description and the Figure 2 legend have been modified accordingly.

  • The presence of cleavage products is a major concern since it is impossible to determine if the expected protein or the cleavage product performs the binding to the receptor. Additional purification was performed? If yes, insert the dates.

Figure 5A has been modified and now present a Western blot with PRC1 (green) and PRC1-Hinge-Fc in red. The full-length PRC1 and PRC1-Hinge-Fc are identified with arrows. The fact that the anti-mouse antibody also revealed the signal comprised between the 43 and the 34 kDa markers demonstrates that it is a fragment of PRC1-Hinge-Fc that contains the mouse Fc. This result suggests the existence of a N-terminal cleavage within the VHH or between the VHH and the Hinge, that generates are smaller form that do not bind to the FSHR. Lines 240-243 have been modified accordingly.

  • Figure 5 A: The figure should contain the molecular weight of all standard bands.

All the standard bands have been annotated in revised Figure 5A.

  • Figure 6: Scales must be the same for the graphs.

Scales have been homogenized in revised Figure 6 as requested.

  • Discussion must be improved; the authors mostly summarize the results.

Two new section on PCR1-Hinge-Fc allosteric binding and on its mode of action has been added to the discussion. Lines 390-407.

  • Minor editing of English language required.

The manuscript has been edited.

Reviewer 2 Report

Comments and Suggestions for Authors

The article describes a phage based method to modulate FSHR and is generally fine, except that there are some areas for improvement.

The title claims the VHH has allosteric modulation activity but the key point here is that allosteric activity is speculation at best and not really demonstrated conclusively. There appears to be gaps in the results that needs to be addressed before the article is suitable for publication.

In the introduction, the orthosteric sites are mentioned as allosteric, this can be further elaborated. While allosteric sites are often outside the the active binding site, they often do have measurable protein backbone effect, and are not limited to VHHs. So the introduction can be expanded to make this clearer.

On the results, some areas of lacking clarity. It starts with the immunization of the Llamas with the HEK293 membranes "or" an expression vector.  The vector was used as an immunizing agent?

More info on the "different forms of the human FSHR" ought to be provided since they are in the selection and screening.

Under section 2.4, VHH-Hinge-Fc in pcDNA3.1 was transfected. The processes were not provided with suitable information. Also, the measurement of the expression and the characterization of the VHH-Hinge-Fc can be shown.

The lack of effect on cAMP production does not conclusively demonstrate  allosteric modulation. This is quite a leap in logic. There are epitopes that the antibody can recognize that do not necessary mean allosteric activity to the active site - this is also a point made in lines 231 - 233.

The VHH can bind outside the active site per se, but due to its location and size, it can block the necessary activation. Such do not demonstrate "allosteric" activity. In fact, the study should map the exact epitope and analyze the epitope for allosteric activity to the active site through bioinformatics. There are such databases that can make the claim of allostery stronger, and is a better and faster alternative than using experimental means. If the authors do not wish to do the allosteric analysis, then the title and writing should not overly state that it is allosterically inhibiting and discuss the possible steric hindrances.

Some references that may be useful to unravel this allosteric part

  • DOI: 10.1016/j.bbapap.2013.01.024
  • https://doi.org/10.1093/abt/tbac005
  •  

Author Response

Reviewer 2’s queries:

  • The title claims the VHH has allosteric modulation activity but the key point here is that allosteric activity is speculation at best and not really demonstrated conclusively. There appears to be gaps in the results that needs to be addressed before the article is suitable for publication.

We replaced “allosteric modulation” in the title by “pharmacologically-active”. We also provided new data that demonstrate no competition between PCR1 or PRC1-Hinge-Fc and FSH for binding to the receptor. This combined with -our other data clearly indicates that PRC1 binds an allosteric site (i.e distinct from the orthosteric site) of FSHR-ECD. However, we agree with this reviewer that demonstrating allosteric activity is a different matter that remains speculative at this stage. Further pharmacological and structural studies would be necessary to demonstrate and decipher the allosteric pathways involved. The revised manuscript has been edited throughout in order to clearly discriminate “allosteric binding” from “allosteric activity or positive allosteric modulation” and to avoid any overstatement on this subject.

  • In the introduction, the orthosteric sites are mentioned as allosteric, this can be further elaborated. While allosteric sites are often outside the the active binding site, they often do have measurable protein backbone effect, and are not limited to VHHs. So the introduction can be expanded to make this clearer.

The corresponding section of the introduction has been modified/expanded accordingly. Lines 53-67.

  • On the results, some areas of lacking clarity. It starts with the immunization of the Llamas with the HEK293 membranes "or" an expression vector.  The vector was used as an immunizing agent?

This section of the results has been clarified in the revised manuscript. Lines 90-94.

  • More info on the "different forms of the human FSHR" ought to be provided since they are in the selection and screening.

More information is provided in the revised version of the manuscript. Lines 101-107.

  • Under section 2.4, VHH-Hinge-Fc in pcDNA3.1 was transfected. The processes were not provided with suitable information. Also, the measurement of the expression and the characterization of the VHH-Hinge-Fc can be shown.

This section has been revised for more clarity and details have been added. Lines 216-224.

As this step was meant as a rapid and cost/time effective secondary screening, the expression level of PRC1-Hinge-Fc was not checked at this stage.

  • The lack of effect on cAMP production does not conclusively demonstrate allosteric modulation. This is quite a leap in logic. There are epitopes that the antibody can recognize that do not necessary mean allosteric activity to the active site - this is also a point made in lines 231 - 233.

We agree, this sentence has been removed from section 2.4.

In order to add more evidence in support of the allosteric binding mode of PRC1, we carried out additional experiments using a home-made NanoBiT assay that we recently implemented in our lab. These data clearly demonstrate that neither PRC1 nor PRC1-Hinge-Fc compete with FSH for binding at the FSHR. We inserted these results in Figure 5D-E. We modified the results (lines 281-284), legend (lines 266-273) and materials and methods (lines 592-600) sections accordingly.

  • The VHH can bind outside the active site per se, but due to its location and size, it can block the necessary activation. Such do not demonstrate "allosteric" activity. In fact, the study should map the exact epitope and analyze the epitope for allosteric activity to the active site through bioinformatics. There are such databases that can make the claim of allostery stronger, and is a better and faster alternative than using experimental means. If the authors do not wish to do the allosteric analysis, then the title and writing should not overly state that it is allosterically inhibiting and discuss the possible steric hindrances.  Some references that may be useful to unravel this allosteric part: DOI: 10.1016/j.bbapap.2013.01.024; https://doi.org/10.1093/abt/tbac005

We agree that these are all valid points. The databases suggested by the reviewer are interesting but would require more time to carry out in-depth analysis of our case. As explained above, the revised version of the manuscript has been amended in order to discriminate allosteric binding from allosteric activity and to avoid overstatement. The references suggested have been cited in a new paragraph of the discussion that is focused and allosteric binding versus allosteric activity and the potential role of VHH-Hinge-Fc bulkiness in the proposed mechanism of action. Lines 385-393.

Round 2

Reviewer 1 Report

Comments and Suggestions for Authors

All issues raised were accepted, and the manuscript progressed with introducing new paragraphs and figures. Therefore, we now recommend its publication.

Comments on the Quality of English Language

Small corrections can be made during editing.

Author Response

  • Minor editing of English language required

The English language of the revised manuscript has been fully edited to improve clarity and fluidity while detecting typos.

Reviewer 2 Report

Comments and Suggestions for Authors

The manuscript is much better now, although there are still some point that need addressing.

I'm not sure we can claim pharmacologically active at this point.

My other concern is that if the PRC1-Hing-Fc expression was not checked, how was its presence even confirmed?

The figures are all too small to be seen once printed. The figures need to be modified and made larger.

Author Response

  • I'm not sure we can claim pharmacologically active at this point.

The title of the manuscript has been changed to: Multiplexed Phage Display, High-Throughput Sequencing, and Functional Assays as a Platform for Identifying Modulatory VHHs Targeting the FSHR.

  • My other concern is that if the PRC1-Hing-Fc expression was not checked, how was its presence even confirmed?

We have added in Figure 4A of the revised version, a Western blot revealed with an anti-mouse Fc antibody conjugated to a fluorescent dye. The presence of a band corresponding to PRC1-Hinge-Fc is clearly detected in the crude cellular extracts. The legend of Figure 4 has been completed accordingly (lines 230-233).

  • The figures are all too small to be seen once printed. The figures need to be modified and made larger.

We have increased the lettering sizes in Figures 2, 3, 4 and 6 of the revised manuscript.